# Shape-memory effect in twisted ferroic nanocomposites

Donghoon Kim[1,9], Minsoo Kim [1,9], Steffen Reidt[2], Hyeon Han [3], Ali Baghizadeh[4], Peng Zeng [4], Hongsoo Choi[5], Josep Puigmartí-Luis[6,7], Morgan Trassin [8], Bradley J. Nelson [1], Xiang-Zhong Chen [1]✉ & Salvador Pané [1]✉

The shape recovery ability of shape-memory alloys vanishes below a critical size (~50 nm), which prevents their practical applications at the nanoscale. In contrast, ferroic materials, even when scaled down to dimensions of a few nanometers, exhibit actuation strain through domain switching, though the generated strain is modest (~1%). Here, we develop freestanding twisted architectures of nanoscale ferroic oxides showing shape-memory effect with a giant recoverable strain (>8%). The twisted geometrical design amplifies the strain generated during ferroelectric domain switching, which cannot be achieved in bulk ceramics or substrate-bonded thin films. The twisted ferroic nanocomposites allow us to overcome the size limitations in traditional shape-memory alloys and open new avenues in engineering large-stroke shape-memory materials for small-scale actuating devices such as nanorobots and artificial muscle fibrils.

The development of nanoscale machines, such as nanoelectromechanical systems, nanorobots, nanoscale aerial vehicles, and injectable miniaturized medical devices, places a pressing demand on nanoscale mechanical actuation architectures and materials. Shape-memory alloys (SMAs) have been widely explored as actuating materials because of their large deformation capability caused by reversible martensitic phase transformations. However, their application at the nanoscale is highly constrained because martensitic phase transformations are suppressed below a critical size (~50 nm)[1,2].

Ferroelastic and ferroelectric oxides maintain their electromechanical response at the nanoscale, even in films of just a few atomic layers[3]. Their electromechanical behavior is associated with changes in their crystalline lattice and ferroic domains. The switching of domains can induce recoverable strains and possibly shape-memory effect, whereas the mechanism is different from the martensitic phase

transformation in SMAs[4,5]. At the nanoscale, the domains can be modulated by substrate-induced strain engineering to improve electromechanical response, yet macroscopic actuating strain barely exceeds 1%[5–7]. Substrate removal could allow further tuning of the ferroic domains by changing boundary conditions and, thus, the exploration of unexpected material properties of the freestanding structures, such as superelasticity and large electromechanical response[8–10].

In this work, we demonstrate shape-memory effect with giant recoverable deformations (>8%) in freestanding architectures of ferroic oxide thin film by amplifying domain switching-induced strains through geometrical twist insertion. Twist insertion has been employed in polymeric coiled fibers and metamaterials to amplify strokes[11–14]. However, it has never been explored in ferroic oxide ceramics, because of their brittle nature in bulk form, or because they

[1]Multi-Scale Robotics Lab, Institute of Robotics and Intelligent Systems, ETH Zurich, Tannenstrasse 3, 8092 Zurich, Switzerland. [2]IBM Research Zurich, Säumerstrasse 4, 8803 Rüschilikon, Switzerland. [3]Max Plank Institute of Microstructure Physics, 06120 Halle (Saale), Germany. [4]The Scientific Center for Optical and Electron Microscopy (ScopeM), ETH Zurich, 8093 Zurich, Switzerland. [5]Department of Robotics & Mechatronics Engineering, DGIST-ETH Microrobotics Research Center, Daegu Gyeong-buk Institute of Science and Technology (DGIST), Daegu, Republic of Korea. [6]Departament de Ciència dels Materials i Química Física, Institut de Química Teòrica i Computacional, University of Barcelona (UB), 08028 Barcelona, Spain. [7]Institució Catalana de Recerca i Estudis Avançats (ICREA); Pg. Lluís Companys 23, Barcelona 08010, Spain. [8]Department of Materials, ETH Zurich, 8093 Zurich, Switzerland. [9]These authors contributed equally: Donghoon Kim, Minsoo Kim. ✉e-mail: chenxian@ethz.ch; vidalp@ethz.ch

are mechanically constrained to the substrate on which they are deposited. Notably, when the crystallite size scales down to the nanoscale, ceramic materials show high strength and large elastic strain endurance[8,9,15–18]. This unique property, together with a bilayer design, enables us to realize predefined architectures that cannot be achieved in the single-layer freestanding approach[3,8]. The twisted architecture was fabricated by releasing patterned $BaTiO_3$ /$CoFe_2O_4$ (BTO/CFO) bilayer thin films from the substrate. Large shape-memory effect was observed through in-situ nanomechanical testing. The twisted architecture has a film thickness of ~ 20 nm, overcoming the size limitation encountered in conventional shape-memory alloys.

## Results and discussion

The twisted architecture was fabricated from BTO/CFO epitaxial bilayer thin films with a thickness of 8 nm and 15 nm, respectively (Fig. 1a). First, the BTO/CFO (001) bilayer was deposited onto a MgO (001) substrate using pulsed laser deposition (Fig. S1). CFO was used to provide interfacial stress because of the large lattice mismatch (4 ~ 5%) between the two layers. The films were patterned into linear stripes of 1 μm wide and 70 μm long, with a tilt of 40° with respect to the [010] axis. After chemical etching of the MgO substrate (detailed information in the Methods section), the thin film stripes were released, and interfacial stress caused the film to roll around the [010] axis to form twisted architectures (Fig. 1b). The BTO layer in the released BTO/CFO membrane exhibited ferroelectric properties with clear domain switching behavior (Fig. S2 and S3). Unlike brittle bulk ceramics, these freestanding twisted architectures exhibited a superelastic behavior[19]. Although some structures were distorted by electrostatic and/or Van der Waals forces between the freestanding film and the substrate (Fig. S4), they recovered their original shape after being mechanically detached from the substrate and exhibited

spring-like behavior when pushed/pulled with a microneedle (Movie S1).

When the tensile stress is sufficiently large, the structure maintained the deformation, as can be seen from the changed helical pitch length in Fig. 1c (recoverable strain calculated from the pitch length change is 26.8% and from the total length change is 8.3%). Interestingly, when the electron beam from the scanning electron microscope (SEM) was focused on the deformed structures, they recovered their initial shape (Fig. 1c, Movie S2), and their high superelasticity was maintained (Fig. S5). This phenomenon is analogous to the conventional shape-memory effect, where deformed structures recover their original shape and mechanical properties via thermomechanical martensitic phase transformations[20,21]. However, the shape recovery in the twisted nanocomposite was triggered by the electron beam, i.e., electrical energy as opposed to thermal energy[10,22,23], which greatly facilitates the application of these structures at the nanoscale where localized thermal stimulation is not possible.

The shape-memory effect was also evaluated by in-situ nano-mechanical tensile tests (Fig. 2a, Fig. S6, and Movie S3). The structure was fully stretched with 9 μm elongation and a maximum tensile force of 1.5 μN. Upon complete unloading, the structure recovered to its original shape. Figure 2b shows the non-linear force-displacement relationship of the twisted architectures obtained during the tensile loading and unloading processes. During the tensile test, there was energy dissipation ($E_{dissipation}$), which can be quantified by the area enclosed by the force-displacement curves. $E_{dissipation}$ increased gradually at a small strain level ($4.08 \times 10^{-3}$ J/cm$^3$ to $2.17 \times 10^{-2}$ J/cm$^3$ during 3.6 μm to 7.1 μm elongation), and increased abruptly when the twisted architecture approached the length limit ($2.17 \times 10^{-2}$ J/cm$^3$ to $1.92 \times 10^{-1}$ J/cm$^3$ during 7.1 μm to 9.1 μm elongation, Fig. 2c). The twisted architecture was tested more than one hundred cycles

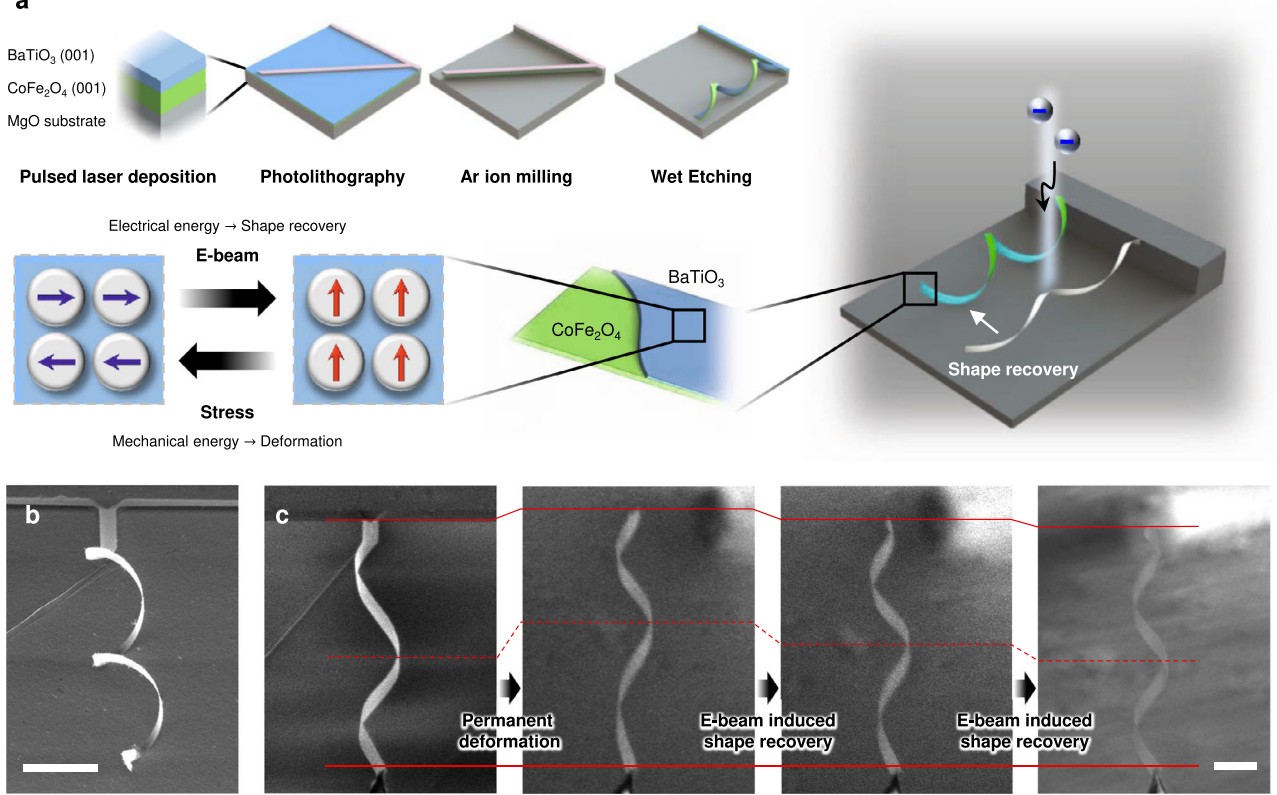

**Fig. 1 | Electron beam induced shape-memory effect in twisted BTO/CFO nanocomposites. a** Schematic diagram of the twisted BTO/CFO fabrication process and the shape-memory effect under electron beam irradiation. **b** SEM image

(45° tilted) of the fabricated twisted BTO/CFO nanocomposite. **c** Permanent deformation after the application of large tensile stress and shape-memory effect under the electron beam irradiation. Scale bars indicate 10 μm.

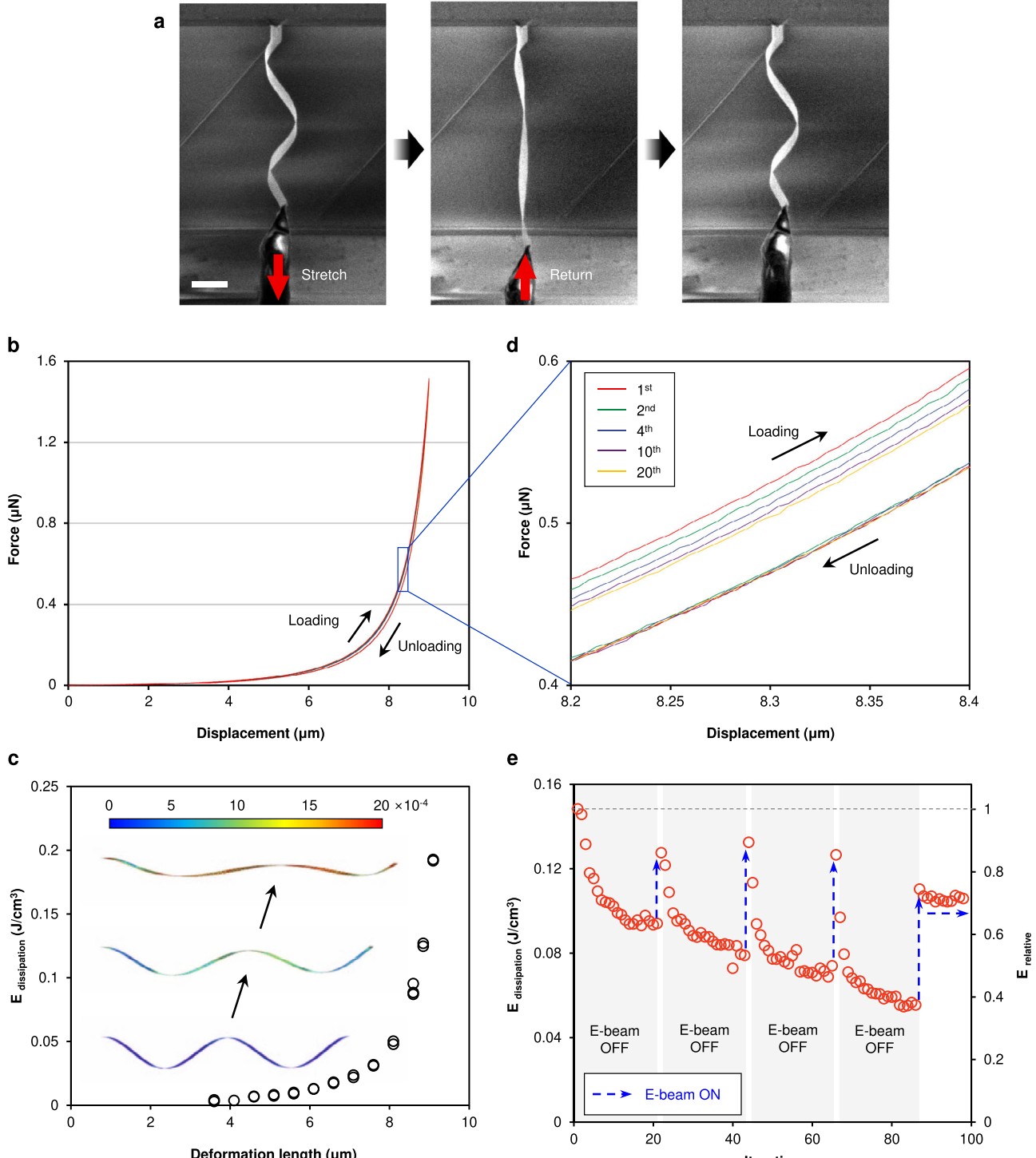

**Fig. 2 | In-situ nanomechanical testing. a** Sequential SEM images of the tensile test of the twisted BTO/CFO (scale bar: 10 μm). **b** Non-linear force-displacement curve measured during tensile loading and unloading. **c** $E_{dissipation}$ as a function of the deformation (stretching) length. Inset images show the strain evolution during stretching, estimated by a finite-element method structural analysis. $E_{dissipation}$ increases exponentially as the twisted architecture is more strained. **d** Magnified force-displacement curves showing the degradation over the repetitions. **e** $E_{dissipation}$ change over the tensile cycling tests. $E_{dissipation}$ degraded over iteration and recovered under electron beam irradiation. Recovered state was maintained under continuous irradiation.

(9 μm stretching) with a gradual decrease in the enclosed areas ($E_{dissipation}$) of the force-displacement hysteresis loop (Fig. 2d, e). Interestingly, $E_{dissipation}$ 'recovered' upon electron beam irradiation (Fig. 2e). After 20 cycles of tensile tests, the structure was re-exposed to the electron beam for a fixed amount of time, and the tensile test was resumed after turning off the electron beam. Notably, $E_{dissipation}$ increased ~ 0.05 J/cm³ after the beam exposure, and this 'recovery' of

the dissipated energy was observed after every re-exposure. When we compared $E_{dissipation}$ during tensile test cycles as a function of the beam current under the continuous beam irradiation condition (Fig. S7), we observed that more energy was consumed with a higher beam current. Additionally, the structure maintained the 'recovered states' (i.e., high-energy-consuming state) under continuous beam irradiation (after 4th cycle in Fig. 2e).

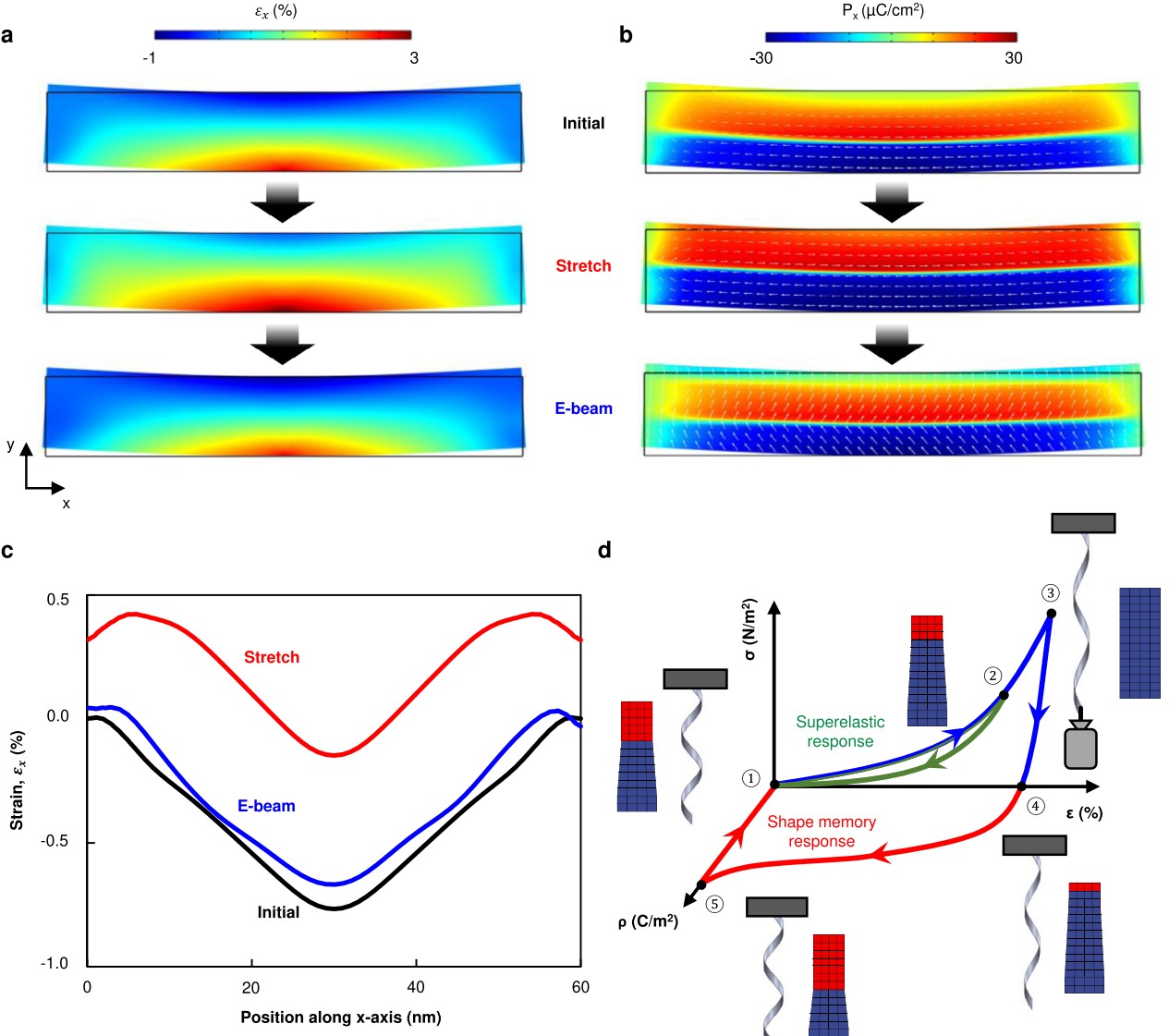

**Fig. 3 | Phase field model simulation on the effect of mechanical stress and electron beam. a** In-plane strain ($\varepsilon_x$) distribution and (**b**) in-plane ferroelectric polarization ($P_x$) mapping in the BTO slab calculated by the phase field model simulation. White arrows indicate the polarization directions. As tensile stress was applied in the BTO slab, the tensile strain and the in-plane domain density increased. The ferroelectric polarizations then switched to out-of-plane directions under surface electrical charging and it caused strain profile go back to the initial state, which can be seen in (**c**) in-plane strain profile of the top of the BTO slab. **d** Proposed cyclic behavior of the superelastic and shape-memory responses with schematic representation of the corresponding ferroelectric domain switching. The blue grids represent in-plane domains, and the red grids represent out-of-plane domains. ρ, σ, and ε indicate surface charge density, stress, and strain, respectively.

From the phenomenological Landau-Ginzburg theory, the elastic strain in ferroelectric materials is expressed as a function of the ferroelectric polarizations:

$$e_{ij} = \varepsilon_{ij} - Q_{ijkl}P_kP_l \qquad (1)$$

where $\varepsilon_{ij}$ is the total local strain and $Q_{ijkl}$ is the electrostrictive coefficient[24]. Therefore, the change of ferroelectric polarization alters the strain in ferroelectric materials and vice versa. Meanwhile, the electron beam irradiation on ferroelectric materials leads to the accumulation of surface charges, which can eventually switch or rotate ferroelectric polarizations to the out-of-plane direction[25–28]. Here, we define the domains with polarization direction in the film plane (or along the tangent of the film plane when the film is curled) as in-plane domains, and those with the polarization direction perpendicular to the tangent of the film plane as out-of-plane domains. As the complex crystalline orientation variations within the twisted architecture make the experimental access to the domain configuration unfeasible, we performed phase-field model simulations (detailed information in the Methods section) based on the time-dependent Landau-Ginzburg equation[29,30]. Figure 3a, b depict the in-plane strain and the ferroelectric domains in the BTO slab of freestanding BTO/CFO, respectively. In the initial state, only the bottom surface was tensile-strained, representing strain coming from the CFO layer, with a domain configuration similar to that of the bent freestanding BTO films[8,31] (top of Fig. 3a, b). When tensile stresses were applied on both sides of the slab (loading tensile force), the slab was mechanically stretched and the in-plane domain density increased (middle of Fig. 3a, b). As negative charges accumulated on the slab surface during the electron beam exposure, ferroelectric polarization switched towards an out-of-plane orientation, giving rise to the change of strain profile and the retraction of the mechanically stretched BTO slab (bottom of Fig. 3a, b, quantitative description of the effect of the tensile force and the surface charge accumulation on the shape recovery is provided in

Fig. S8). The in-plane strain profile along the top surface of the BTO slab shows more clearly the increase of the tensile strain under stretching and the recovery under the change in the charge boundary condition (electron beam irradiation, Fig. 3c). Therefore, it is likely that the switching between the in-plane and out-of-plane oriented ferroelectric domains, induced by mechanical stress and the electric field, is responsible for the observed shape deformation and its recovery.

Based on our findings, we suggest a possible mechanism for superelasticity and the shape-memory effect in the twisted architectures (Fig. 3d). As an epitaxially strained ferroelectric material, BTO possesses a complex domain configuration, where domain switching enables recoverable strains[4,8,32,33]. Shape deformations and recoveries in the BTO/CFO twist were achieved by the interplay between the stress induced by the ferroelectric domain switching in the BTO layer and the mechanical stress imposed by the bottom CFO layer, in contrast to the single BTO membrane where folding-unfolding is achieved by the electrostatic interaction induced by the polarization switching[10]. In the low-strain regime, the twist shows a superelastic response (①→②→① in Fig. 3d). While surface tension-modulated elastic deformation cannot be ruled out[34], ferroelectric polarization switching undoubtedly contributes to the superelastic response. The ferroelectric polarization switches to an in-plane direction during stretching and returns to the original direction during release, as shown by the simulation results and as observed elsewhere[8]. The deformation of the twist is maintained in the large strain regime (beyond the threshold strain) even after the removal of the external stress (②→③→④ in Fig. 3d), which can be attributed to either the residual stress or domain pinning during mechanical loading[35–37]. This differentiates mechanical responses in the BTO/CFO twist from the electromechanical effect in single BTO membrane where continuous external stimulus is required to maintain the deformation[10]. When the electron beam is focused on the structure, the ferroelectric polarization switches from in-plane to out-of-plane, providing the force for shape recovery (④→⑤→① in Fig. 3d). The in-plane domain pinning can be corroborated by the gradual decrease in $E_{dissipation}$ with an increased cycling number (Fig. 2e), although the movement and/or accumulation of the defects, such as dislocations in the structure, could also contribute to the decrease of $E_{dissipation}$[38,39]. We tested the repeatability of this shape-memory response. Regardless of the type of external force that was used for deformations, the shape-memory effect was consistently observed with electron beam irradiation (Movies S4, S5).

The recovery of the $E_{dissipation}$ after the beam exposure and the beam dose dependency of the $E_{dissipation}$ clearly demonstrates the involvement of the polarization-switching modulated mechanism (Fig. S7). The other possible effects, such as heating, electrostatic charging, or vibration, can be ruled out. As can be seen in Fig. 2e, $E_{dissipation}$ does not depend on the exposure time and maintains constant value under the continuous electron beam exposure (after 4th cycle), indicating the shape-memory effect is not induced by the heating[10,40]. In addition, the actuation force measured with BTO/CFO twist proves that the electrostatic charging and vibration effect are negligible (Fig. S9).

Still, the effects of the magnetostriction and/or the magnetic domain configurations in the CFO layer remain elusive. Other than imposing mechanical constraints in the BTO layer, the CFO layer could play a role in the shape-memory effect through the magnetostriction or the change of the magnetic domain configurations. However, we have not observed any shape recovery and $E_{dissipation}$ values were also unaffected in the presence of magnetic fields, up to 50 mT (Fig. S10 and Movie S6).

Compared to conventional SMAs where martensitic phase transformation is suppressed below the critical size, a domain switching-induced shape-memory effect in ferroic oxides was observed in films of ~20 nm (Fig. 4a). This is the smallest feature size at which shape-memory effect has been demonstrated. Our results can also be

extrapolated to smaller scales, as ferroelectricity occurs even at the monolayer[3]. In addition, the electric field-driven shape recovery of twisted architecture shows remarkable recoverable strains (Fig. 4b) as well as the clear actuation force (Fig. S9), providing opportunities to develop new type of the small-scale electromechanical actuating systems. The twisted architectures may also have the potential to be actuated with other external stimuli, such as temperature and magnetic fields, since (i) BTO goes through phase transformations at Curie temperatures, and (ii) CFO is ferromagnetic, which may in turn broaden their applications.

In summary, we designed and demonstrated ferroelectric domain switching-induced shape-memory twisted architectures by changing the boundary condition via releasing the film from the substrate and introducing geometric engineering. An electrically induced large recoverable strain was achieved in the twisted architectures. Shape-memory effects realized through this approach can bypass the critical size limitation encountered in the conventional SMAs. We believe that our discovery will enable new large-stroke shape-memory materials and structures for small-scale devices, such as micro- and nanorobots, actuators, and artificial muscles.

## Methods

### Thin film deposition
BTO/CFO epitaxial thin films were grown on (001) oriented MgO single crystalline substrate (Crystal GmbH) using pulsed laser deposition with a 248 nm KrF excimer laser, as previously reported[41]. MgO substrate was cleaned with acetone and ethanol under ultrasonic wave for 5 min, respectively, before the deposition process. A CFO layer was first deposited at 550 °C, 10 mTorr oxygen partial pressure, and laser parameters of 5 Hz, 1.8 J/cm$^2$. Subsequently, a BTO layer was deposited at 750 °C, 200 mTorr oxygen partial pressure with 4 Hz, 1.2 J/cm$^2$ laser. Theta-2theta X-ray diffraction and reciprocal space mappings show the epitaxial nature of as-deposited thin films with relaxed strain status of the BTO layer (Fig. S1).

### Twisted BTO/CFO fabrication
Photoresist (AZ 1505) was spin-coated onto the BTO/CFO thin films, and arrays of tilted lines with 1 μm width, 70 μm length, and 40° angle from [010] axis were patterned using UV-photolithography on the film. Subsequently, the patterned film was dry-etched with Ar-ion milling (Oxford IonFab 300 Plus) with a 500−mA beam current for 20 s, for a total 15 times with 90 s resting between each milling session to prevent the photoresist from burning. After the dry-etching, the remaining photoresist was rinsed-off with acetone (5 min), IPA (5 min), and oxygen plasma (600 W, 3 min). Afterwards, the MgO substrate was wet-etched with sodiumbicarbonate saturated solution[42] and the sample was dried with a critical point dryer (Tousimis-CPD).

### In-situ nanomechanical tensile testing
Tensile tests were performed under a scanning electron microscope (Nova NanoSEM 450, FEI Company) equipped with a nanomechanical testing system (FT-NMT03, Femtotools AG) at room temperature (Fig. S4). Twisted BTO/CFO nanocomposites were attached to the force sensor probe using SEM-compatible glue (SEMGLU, Kleindiek Nanotechnik GmbH), and the force was measured with 100 Hz sampling frequency under a 0.5 μm/s tensile loading-unloading rate using a micro-electro-mechanical system (MEMS)-based force sensor (model FT-S200). The force sensor has a tungsten probe tip with a radius of <0.1 μm, ±200 μN force range limit, and 0.5 nN resolution.

### Ferroelectric property characterizations of the twisted BTO/CFO
Ferroelectric properties of the twisted BTO/CFO were measured with piezoresponse force microscopy (PFM, NT-MDT). Freestanding BTO/

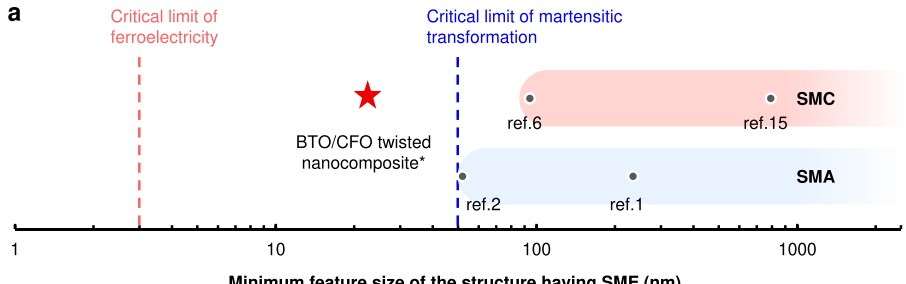

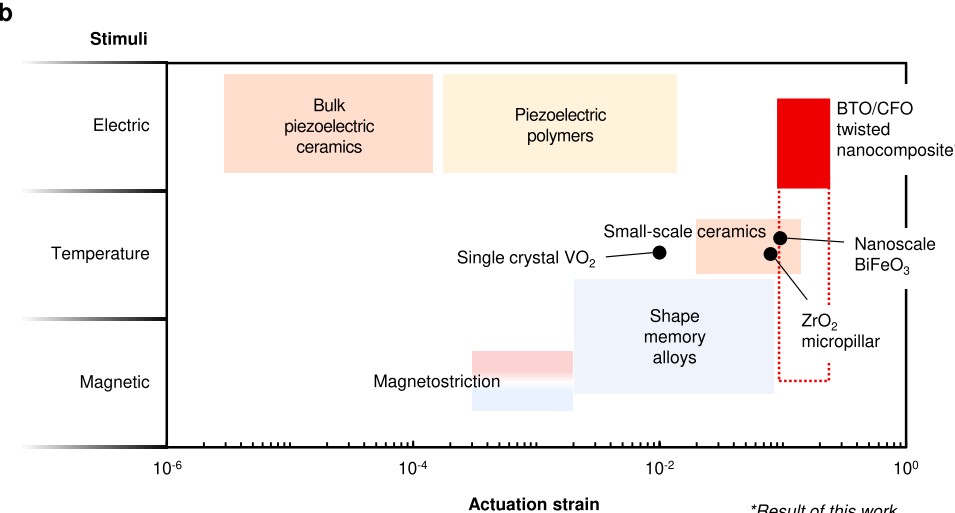

**Fig. 4 | Comparison of feature size and actuation strain in different materials.** **a** The minimum feature size of materials with shape-memory effect (SME). For martensitic phase transformation-based shape-memory alloys (SMAs) and ceramics (SMCs), the critical feature size is limited to ~50 nm as the phase transformation is suppressed below this limit. For shape-memory ferroelectrics, however, ferroelectric domain switching can occur down to a few nanometers, extending the minimum feature size to a smaller scale. **b** Comparison of the actuation strain in different materials grouped by the actuation stimulus (Red: ceramics, blue: metals, yellow: polymers). In a BTO/CFO twisted nanocomposite, the shape-recovery actuation can achieve >10%. Although the recoverable strain from the ferroelectric domain switching in the BTO is around 1%, the shape-memory effect is amplified by the structural design, giving a large actuation strain range.

CFO membranes were transferred onto Au-coated Si substrate and ferroelectric polarization switching was measured by applying ± 10 V voltages. For atomic structure analysis, TEM lamellae of the freestanding BTO/CFO layer were prepared by Focused Ion Beam (FIB, The Thermo Scientific Helios 5 UX) and atomic displacements of Ti-ions in BTO layer and the ferroelectric domain configuration were analyzed with high-angle annular dark field scanning transmission electron microscopy (HAADF-STEM, double aberration-corrected JEOL GrandARM operated at 200 kV). Scan-distortion compensation was done using the SmartAlign software by taking 30 frames of STEM images and by applying the Rigid Alignment[43]. High-frequency noises were removed by applying two-dimensional Wiener filter using HREM-Filters Pro software. The displacement of the Ti-ions with respect to the Ba cage was calculated from Atomap[44] and TEMUL[45] open-source jupyter-based packages. The noise reduction on the calculated polarization vectors was done by applying principal component analysis (PCA) implemented in the Scikit-learn library of python.

**Phase-field modeling**
Ferroelectric polarization switching behavior was investigated using phase-field modeling. For simplicity, two-dimensional BTO slabs (60 nm × 10 nm) with different mechanical and charge boundary conditions were simulated by solving the time-dependent Ginzburg-Landau equation,

$$\frac{\partial P_i(r,t)}{\partial t} = -L\frac{\delta F}{\delta P_i(r,t)}, i = 1,2,3 \qquad (2)$$

where $P_i(r,t)$ is the polarization at location $r$ and time $t$, $L$ is a domain wall mobility related kinetic coefficient, and $F$ is the total free energy that can be expressed as

$$F = \iiint \left(f_{bulk} + f_{elec} + f_{grad} + f_{elas}\right) dV \qquad (3)$$

and the Landau free-energy density ($f_{bulk}$), electric energy density ($f_{elec}$), gradient energy density ($f_{grad}$), and elastic energy density ($f_{elas}$), are described by

$$f_{bulk} = \alpha_{ij}P_iP_j + \alpha_{ijkj}P_iP_jP_kP_l + \alpha_{ijkjmm}P_iP_jP_kP_lP_mP_n \qquad (4)$$

$$f_{elec} = -\frac{1}{2}\varepsilon_0\varepsilon_b E_i E_j - E_i P_i \qquad (5)$$

$$f_{grad} = \frac{1}{2}G_{ijkl}P_{i,j}P_{k,l} \qquad (6)$$

$$f_{elas} = \frac{1}{2}C_{ijkl}\left(\varepsilon_{ij} - \varepsilon_{ij}^o\right)\left(\varepsilon_{kl} - \varepsilon_{kl}^o\right) \qquad (7)$$

where $\alpha$'s are the Landau expansion coefficients, $\varepsilon_0$ and $\varepsilon_b$ are the vacuum permittivity and dielectric constant, $E_i$ is the electric field including both external field and depolarization field, $G_{ijkl}$ is the gradient energy coefficient, $P_{i,j}$ is the polarization gradient, $C_{ijkl}$ is the elastic stiffness tensor, and $\varepsilon_{ij}$ and $\varepsilon_{ij}^o$ are the total and spontaneous

strain, respectively. The coefficients for the equations were adopted from the previous research[46,47]. Equation (2) was transformed into general partial differential equation forms in finite-element method software COMSOL Multiphysics and numerically solved with 1 nm × 1 nm mesh size. Surface tension (which represents the interfacial strain from the CFO at the bottom surface of the BTO slab) and boundary loads (tensile forces on both sides of the slab) were considered as mechanical boundary conditions and a closed loop electrical boundary condition was adopted.

## Reporting summary

Further information on research design is available in the Nature Portfolio Reporting Summary linked to this article.

## Data availability

All data are available in the main text or the supplementary materials.

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

## Acknowledgements

This work has been financed by the ERC Consolidator Grant "Highly Integrated Nanoscale Robots for Targeted Delivery to the Central Nervous System" HINBOTS under the grant no. 771565, the MSCA-ITN training program "mCBEEs" under the grant no. 764977, the ERC Advanced Grant "Soft Micro Robotics" SOMBOT under the grant no. 743217, and the Swiss National Science Foundation (Project No. 200021L_192012). X. C. would like to acknowledge the Swiss National Science Foundation (No. CRSK-2_190451) for partial financial support. This work was partially supported by the National Research Foundation of Korea (NRF) (No. 2021M3F7A1082275 and No. 2017K1A1A2013237), funded by the Ministry of Science and ICT of Korea. M. T. acknowledges the financial support by the Swiss National Science Foundation under project No. 200021_188414. M. K. acknowledges partial financial support by the Swiss National Science Foundation under project No. 200021L_197017.The authors would also like to thank the Scientific Center for Optical and Electron Microscopy (ScopeM) and the FIRST laboratory at ETH for their technical support, and the Cleanroom Operations Team of the Binning and Rohrer Nanotechnology Center (BRNC) for their help and support. The authors appreciate Dr. Marta D. Rossell in Empa for her help and support in preliminary microstructural analysis. X.C. also would like to thank Dr. Ming Chen and Dr. Xin Chen for useful discussions.

## Author contributions

D.K., M.K., and X.C. proposed and designed the project. D.K. and M.K. performed the experiments, analyzed the data, and developed the theoretical model. S.R. and H.H. contributed to the material characterization. A.B. and P.Z. performed the TEM experiments and analyzed the data. H.C., J.P.-L., and M.T. contributed to discussion. D.K., M.K., M.T., and X.C. wrote the manuscript with contributions from all authors. B.J.N., X.C., and S.P. supervised the work.

## Competing interests

The authors declare no competing interests.
