## [Peer Review File · Nature Communications]

Shape-memory effect in twisted ferroic nanocompositesREVIEWER COMMENTS

Reviewer #1 (Remarks to the Author):

In this manuscript, the authors report a shape-memory effect with giant recoverable deformations (>10%) in freestanding twisted architectures of ferroic oxide thin film BaTiO₃/CoFe₂O₄). The authors believe that the twisted geometrical design amplifies the strain generated during ferroelectric domain switching, which overcomes the size limitations in traditional shape-memory alloys. To demonstrate the application of these structures, the authors (i) performed in-situ nanomechanical tensile tests, (ii) applied the electron beam from the scanning electron microscope, and (iii) conducted phase-field simulations. In the end, based on the experimental and simulation findings, they proposed a possible mechanism for superelasticity and the shape-memory effect in their twisted architectures.

The experimental data are presented in sufficient detail and are technically sound. The shape-memory effect in free-standing BaTiO₃ crystal membranes was already reported in other works [Science 366, 475-479 (2019); ACS Nano 2020, 14, 4, 5053–5060]. The twisted insertion is the first time introduced to the free-standing BaTiO₃ in this manuscript as far as the reviewer's knowledge. These results could be of potential significance in engineering functional materials that take advantage of electromechanical response. The paper is also very well written and clear. Hence, this paper is worth to be published in Nature Communications.

I only have one medium comment to give to the authors:

Major remarks:

None.

Medium remarks:

Similar effects were reported in free-standing BaTiO₃ [Science 366, 475-479 (2019); ACS Nano 2020, 14, 4, 5053–5060]. So naively, I would deduce that BaTiO₃/CoFe₂O₄ twisted architectures should share similar explanations. However, the ACS Nano paper reported and explained that single crystal free-standing BaTiO₃ (without epitaxial strain) also exhibits folding (without electron injection) and unfolding (with electron injection) behavior. In my opinion, it is unclear how these two experiments' explanations are connected. Though I also agree that the situation in a twisted structure could be much more complicated, the single crystal free-standing BaTiO₃ is simpler to analyze. I would suggest the authors give more comments on the explanations in these single crystal free-standing BaTiO₃ papers, e.g., why the epitaxial strain is needed to explain the shape-memory effect.

In addition, CoFe₂O₄ is known to be used for its magnetostrictive applications. So, could magnetostriction in CoFe₂O₄ play a role in this twisted structure? Especially when the CoFe₂O₄ is thicker than the BaTiO₃ layer.

Minor remarks:

None

Some possible typos:

1. "The twisted ferroic nanocomposites allow us to overcome the size limitations in traditional shape-memory alloys and *opens* new avenues"

opens -> open

2. "The structure was fully stretched with 9 μm elongation and a *maximun* tensile force of 1.5 μN ."

maximum => maximum

We tested the repeatability of this shape-memory response and irrespective of the type of external force that was used for deformations*, * the shape-memory effect was consistently observed with electron beam irradiation (movies S4, S5).

, => ;

3. "the critical feature size is limited to *apporoximately* 50 nm"

apporoximately -> approximately

Reviewer #2 (Remarks to the Author):

Report for "Giant shape-memory effect in twisted ferroic nanocomposites" by Donghoon Kim, et al.

The authors demonstrated an investigation of giant shape-memory effect induced by electron beam in free-standing twisted BTO/CFO bilayer thin films. In-situ nanomechanical testing and phase-field model simulations were combined to clarify the effect of mechanical stress and electron beam. However, I do not think that the scientific quality and expression are good enough for the high impact journal of Nature Communications, therefore I suggest this manuscript should be rejected. Besides, a similar research work has been recently published in Adv. Mater. 2022, 34 (13), 2108419. In addition, the obvious drawbacks of this manuscript can be seen:

1: The recoverable strain > 10% mentioned in Abstract does not have a corresponding quantitative result in the main text.

2: The statement of "experimental access to the domain configuration unfeasible" lacks of factual basis, for example, the FIB-SEM has been used to reveal the domain configurations in BTO (10.1002/adma.202270103). Without the necessary microstructural characterization, the referee doubts the reliability of the followed simulation results. And, the influence of the defects and contamination introduced by the sample preparation on the ferroelectric domain switching are not clear.

3: The electron beam irradiation leded to heating, electrostatic charging and vibration etc., it is unreasonable to directly attribute the increase of Edissipation to the irradiation-induced polarization. As shown in the SI videos, there was a significant charging at the top of the tip, and the effect of the complex electrostatic field on the beam induced-deformation is not clear too.

4: Phase-field model simulation was tried to unveil the physical mechanism, in which was a BTO/BTO structure. It is not consistent with the experimental BTO/CFO structure. The simulation results were too crude because there was not any quantitative result about the strain induced by the polarization. Therefore, I am confused about how the twisted geometrical design in the BTO/CFO bilayered thin films amplified the strain.

Reviewer #3 (Remarks to the Author):

In this work, the authors discover that the bilayer nanostructured BaTiO₃/CoFe₂O₄ coils possess superelasticity. Furthermore, they found the materials exhibit shape memory effect under electron irradiation after the coils are stretched to a high strain. A polarization-switching mechanism of ferroelectric layer is proposed to explain the superelasticity and shape memory effect. Overall, the results are quite interesting and publishable in NC although the shape memory effect and superelasticity have been reported in bulk and thin film ferroelectrics. I hope the authors could consider the following issues before the work can be accepted for publication:

- (1) The work lacks microstructure characterization. The authors attribute to the observed superelasticity and shape memory effect to the polarization-switching process. It implies that BTO is in the ferroelectric state. The authors may provide more results showing BTO is indeed in ferroelectric state. Also, if possible, the direct observation of the polarization or domain switching is also important.
- (2) The film is composed of BaTiO₃ and CoFe₂O₄. The authors simply attribute the observed superelasticity and shape memory effect to the ferroelectric layer. Does CoFe₂O₄, which is also a ferroic material, contribute to the observe phenomena in the nanostructure? If not, evidence or discussion could be provided.
- (3) From the results in the manuscript, shape memory is a real effect of the nanostructure. The elastic deformation of the coils is the amplified one due to the special structure of the material. The real strain of the material might not be large and be termed as superelasticity. The authors need prove that the superelasticity is also the property of the material.
- (4) The title is recommended to be "Shape-memory effect in twisted BaTiO₃/CoFe₂O₄ ferroic nanocomposites" because I am not sure the exact meaning of the word "giant" ("giant" strain?).

Responses to Reviewers' Comments

Reviewer 1:

General comments

In this manuscript, the authors report a shape-memory effect with giant recoverable deformations (>10%) in freestanding twisted architectures of ferroic oxide thin film ($\text{BaTiO}_3/\text{CoFe}_2\text{O}_4$). The authors believe that the twisted geometrical design amplifies the strain generated during ferroelectric domain switching, which overcomes the size limitations in traditional shape-memory alloys. To demonstrate the application of these structures, the authors (i) performed in-situ nanomechanical tensile tests, (ii) applied the electron beam from the scanning electron microscope, and (iii) conducted phase-field simulations. In the end, based on the experimental and simulation findings, they proposed a possible mechanism for superelasticity and the shape-memory effect in their twisted architectures.

The experimental data are presented in sufficient detail and are technically sound. The shape-memory effect in free-standing BaTiO_3 crystal membranes was already reported in other works [Science 366, 475-479 (2019); ACS Nano 2020, 14, 4, 5053–5060]. The twisted insertion is the first time introduced to the free-standing BaTiO_3 in this manuscript as far as the reviewer's knowledge. These results could be of potential significance in engineering functional materials that take advantage of electromechanical response. The paper is also very well written and clear. Hence, this paper is worth to be published in Nature Communications.

Response:

We are thankful for the reviewer's careful review of our work with the endorsement “These results could be of potential significance in engineering functional materials that take advantage of electromechanical response”, and we appreciate his/her suggestive comments. However, we have a slightly different opinion on this sentence “The shape-memory effect in free-standing BaTiO_3 crystal membranes was already reported in other works...”, as we believe that no shape memory phenomenon but only super-elasticity and giant electromechanical response were mentioned in these papers. Based on the constructive comments that the reviewer provided, we have revised our work as follows.

Comment 1: Similar effects were reported in free-standing BaTiO_3 [Science 366, 475-479 (2019); ACS Nano 2020, 14, 4, 5053–5060]. So naively, I would deduce that $\text{BaTiO}_3/\text{CoFe}_2\text{O}_4$ twisted architectures should share similar explanations. However, the ACS Nano paper reported and explained that single crystal free-standing BaTiO_3 (without epitaxial strain) also exhibits folding (without electron injection) and unfolding (with electron injection) behavior. In my opinion, it is unclear how these two experiments' explanations are connected. Though I also agree that the situation in a twisted structure could be much more complicated, the single crystal free-standing BaTiO_3 is simpler to analyze. I would suggest the authors give more comments on the explanations in these single crystal free-standing BaTiO_3 papers, e.g., why the epitaxial strain is needed to explain the shape-memory effect.

Response:

We are grateful to the reviewer for pointing out the importance of clarifying the similarities and differences between the single crystal free-standing BaTiO₃ and the twisted BaTiO₃/CoFe₂O₄. As the reviewer mentioned, H. Elangovan *et al.* reported the folding and unfolding of the single crystal free-standing BaTiO₃ under the electron injection (ACS Nano 2020, 14, 5053-5060). While both BaTiO₃ membrane and twisted BaTiO₃/CoFe₂O₄ show shape recovery, there are major differences in mechanisms.

In free-standing BaTiO₃ membranes, the reversible folding-unfolding (“sushi-rolling-like motion”) is originated from the electrostatic interactions. Opposite directions of ferroelectric domains in a BaTiO₃ membrane align periodically, resulting in the formation of surface charges with opposite polarity at the top and the bottom parts of the membrane. These charged surfaces attract each other and the electrostatic attraction acts as the driving force for the rolling in the natural state. When electron beam is exposed, ferroelectric polarization switches and changes electrostatic interactions between the top and the bottom parts, causing ‘unfolding’ of the BaTiO₃ membrane.

In contrast, in twisted BaTiO₃/CoFe₂O₄, the shape memory effect is originated from the stress equilibrium. The bottom CoFe₂O₄ layer imposes interfacial mechanical stress to the BaTiO₃ layer because of the lattice mismatch (~5 %), which causes the rolling-up of the bilayer and leads towards the twisted architectures. When external tensile stress is applied, the ferroelectric polarization in the BaTiO₃ layer switches to the in-plane direction for the stress equilibrium. When the electron beams are exposed, ferroelectric polarizations switch back to the out-of-plane direction, recovering the original stress equilibrium state and thus the original shape (Fig. 3).

Another difference between the two phenomena is the continuity and spontaneity of the electromechanical responses. In single BaTiO₃ membrane, the membrane unfolds continuously with the higher electron beam and as soon as the beam is turned off, the ‘unfolded’ BaTiO₃ membrane folds back spontaneously, recovering its original state. In this single BaTiO₃ membrane, maintaining ‘unfolded’ state is not possible without the electron beam exposure. On the other hand, in BaTiO₃/CoFe₂O₄ twists, after applying tensile stress over the threshold, the twisted architecture permanently deforms and maintains the deformation even after removal of the external stress (Fig. 1c). As soon as the electron beam is focused on to the twisted structure, the structure recovers its original shape and maintains the original shape. No further structural change was observed after recovering the original shape even with a higher electron beam dosage, because the interfacial strain from CFO bottom layer balances and holds the entire structures.

Based on the reviewer's comment, we have added the detailed explanation comparing similarities and differences between the single crystal free-standing BaTiO₃ membrane and the twisted BaTiO₃/CoFe₂O₄ in the revised manuscript.

Revisions to the manuscript:

“Shape deformations and recoveries in the BTO/CFO twist were achieved by the interplay between the stress induced by the ferroelectric domain switching in the BTO layer and the mechanical stress imposed by the

bottom CFO layer, in contrast to the single BTO membrane where folding-unfolding is achieved by the electrostatic interaction induced by the polarization switching¹⁰. In the low-strain regime, the twist shows a superelastic response (①→②→① in Fig. 3d). While surface tension-modulated elastic deformation cannot be ruled out³², ferroelectric polarization switching undoubtedly contributes to the superelastic response. The ferroelectric polarization switches to an in-plane direction during stretching and returns to the original direction during release, as shown by the simulation results and as observed elsewhere⁸. The deformation of the twist is maintained in the large strain regime (beyond the threshold strain) even after the removal of the external stress (②→③→④ in Fig. 3d), which can be attributed to either the residual stress or domain pinning during mechanical loading³³⁻³⁵. This differentiates mechanical responses in the BTO/CFO twist from the electromechanical effect in single BTO membrane where continuous external stimulus is required to maintain the deformation¹⁰.”

Fig. 1. Electron beam induced shape-memory effect in twisted BTO/CFO nanocomposites. (a) Schematic diagram of the twisted BTO/CFO fabrication process and the shape-memory effect under electron beam irradiation. (b) SEM image (45° tilted) of the fabricated twisted BTO/CFO nanocomposite. (c) *Permanent deformation after the application of large tensile stress and shape-memory effect under the electron beam irradiation.* Scale bars indicate 10 μm.

Comment 2: In addition, CoFe₂O₄ is known to be used for its magnetostrictive applications. So, could magnetostriction in CoFe₂O₄ play a role in this twisted structure? Especially when the CoFe₂O₄ is thicker than the BaTiO₃ layer.

Response:

We appreciate for the reviewer's comment on the role of magnetostriction of CoFe₂O₄ (CFO). It is well-known that the magnetic shape memory effect is originated from the magnetic field induced phase

transformations that involve movements of the twin boundaries (ECS Transactions, 2007, 3 (25), 155-163, Adv. Func. Mat., 2009, 19, 983-998). However, CFO doesn't go through such phase transformation under magnetic fields, but only shows ferromagnetic and magnetostrictive properties (max. 600 ppm, Phys. Rev., 1955, 99, 1788). We believe that the magnetostriction of CFO itself is not strong enough to cause any shape memory effect in twisted BTO/CFO architectures based on our following experimental results.

Indeed, our CFO layer shows ferromagnetic behavior and its magnetic domain will react under the application of the magnetic fields (Fig. R1 a). In order to investigate the effect of the magnetostriction and/or the effect of the magnetic domain configuration changes in the CFO layer, we performed nanomechanical tensile tests under the external magnetic fields. To exclude the effect of the electron beam and the vibration as well as to apply magnetic fields, the setup was built on a vibration-isolated table equipped with optical microscope (Fig. R1 b and Movie S6). Structural deformation of the twist has been observed under large tensile stress applications; however, we have not observed any shape recovery with the magnetic field applications, up to 50 mT. In addition, $E_{\text{dissipation}}$ values during the tensile test cycles were also unaffected in the presence of magnetic fields (Fig. R1 c).

Fig. R1. Magnetic field dependency of the mechanical properties of BTO/CFO microhelices. (a) Magnetic hysteresis loop of the BTO/CFO/MgO (001) thin film. (b) In-situ nanomechanical measurement setup under optical microscope with magnetic field application system. The tensile test under optical microscope is available in Movie S6. Magnetic field was calibrated with magnetometer before applying the field. (c) Effect of the magnetic field (~ 50 mT) on the $E_{\text{dissipation}}$ over the tensile cycling tests. Unlike electron beam irradiation, no significant recovery was observed.

Based on the results above, we revised the manuscript as following:

Revisions to the manuscript:

“Still, the effects of the magnetostriction and/or the magnetic domain configurations in the CFO layer remain elusive. Other than imposing mechanical constraints in the BTO layer, the CFO layer could play a role in the shape memory effect through the magnetostriction or the change of the magnetic domain configurations. However, we have not observed any shape recovery, and $E_{dissipation}$ values were also unaffected in the presence of magnetic fields, up to 50 mT (Fig. S10 and Movie S6).”

Fig. S10. Magnetic field dependency of the mechanical properties of BTO/CFO microhelices. (a) Magnetic hysteresis loop of the BTO/CFO/MgO (001) thin film. (b) In-situ nanomechanical measurement setup under optical microscope with magnetic field application system. The tensile test under optical microscope is available in Movie S6. Magnetic field was calibrated with magnetometer before applying the field. (c) Effect of the magnetic field (~50 mT) on the $E_{dissipation}$ over the tensile cycling tests. Unlike electron beam irradiation, no significant recovery was observed.

Comment 3:

Some possible typos:

1. “The twisted ferroic nanocomposites allow us to overcome the size limitations in traditional shape-memory alloys and *opens* new avenues”
opens -> open
2. “The structure was fully stretched with 9 μm elongation and a *maximun* tensile force of 1.5 μN .”
maximum -> maximum
3. We tested the repeatability of this shape-memory response and irrespective of the type of external force that was used for deformations*,* the shape-memory effect was consistently observed with electron beam irradiation (movies S4, S5).
, -> ;
4. “the critical feature size is limited to *apporoximately* 50 nm”
apporoximately -> approximately

Response:

We are grateful for the reviewer's comment. Based on the comment, we have revised our manuscript.

Revisions to the manuscript:

*“The twisted ferroic nanocomposites allow us to overcome the size limitations in traditional shape-memory alloys and **open** new avenues in engineering large-stroke shape-memory materials for small-scale actuating devices such as nanorobots and artificial muscle fibrils.”*

*“The structure was fully stretched with 9 μm elongation and a **maximum** tensile force of 1.5 μN .”*

*“We tested the repeatability of this shape-memory response. **Regardless** of the type of external force that was used for deformations, the shape-memory effect was consistently observed with electron beam irradiation (movies S4, S5).”*

*“For martensitic phase transformation-based shape-memory alloys (SMAs) and ceramics (SMCs), the critical feature size is limited to **approximately** 50 nm as the phase transformation is suppressed below this limit.”*

Reviewer 2:

General Comments:

The authors demonstrated an investigation of giant shape-memory effect induced by electron beam in free-standing twisted BTO/CFO bilayer thin films. In-situ nanomechanical testing and phase-field model simulations were combined to clarify the effect of mechanical stress and electron beam. However, I do not think that the scientific quality and expression are good enough for the high impact journal of Nature Communications, therefore I suggest this manuscript should be rejected. Besides, a similar research work has been recently published in *Adv. Mater.* 2022, 34 (13), 2108419.

Response:

We are thankful for the reviewer's efforts in reviewing our work. However, we politely disagree with the reviewer's opinion that our work is similar to the previously published paper (*Adv. Mater.* 2022, 34 (13), 2108419). It is undeniable that is a nice piece of work, and the results in that work and other work from the same group (e.g., *Science* 2019, 366, 475-479, *Appl. Phys. Lett.*, 2020, 116, 152903) can help us explain certain phenomena (e.g., the domain arrangement at the initial state) in our work. However, we have to stress again that in our current work, we reported a distinct phenomenon: e-beam induced shape memory effect, which has fundamentally different approaches from the previously reported work. Our work has several major differences from the paper in many aspects:

1. We report the electron beam induced shape memory effect with giant recoverable strain in BTO/CFO ferroic nanocomposites with the twisted architecture for the first time by demonstrating shape deformation and recovery under electron beam and by analyzing quantitative nanomechanical test results. On the contrary, in the paper mentioned by the reviewer (*Adv. Mater.* 2022, 34 (13), 2108419), only large compressibility and stretchability of 'nanospring' are demonstrated without reporting any shape memory effect nor thorough quantitative analysis of mechanical properties.
2. We provide thorough mechanical characterization of BTO/CFO twists by performing advanced nanomechanical tensile tests, including (i) force-displacement hysteresis loops measured during tensile tests over 100 cycles, (ii) deformation length dependency of the force-displacement curves and the energy dissipation of the hysteresis loops, (iii) recovery of the energy dissipation by the electron beam irradiation, and (iv) the electron beam intensity dependency of the energy dissipation. Along with the SEM images and videos, these quantitative and thorough mechanical analysis further corroborates our claim that BTO/CFO twists have shape memory effect under the electron beam irradiation.
3. We also show the repeatability of the shape memory effect in BTO/CFO twists. Different external stimuli (direct mechanical stretch, using Van der Waals force, and using electrostatic interaction) have been employed to give permanent mechanical deformation in BTO/CFO twists. After the

permanent deformation, the shape memory effect was induced by the electron beam exposure for all cases (Movie S2, S4 and S5).

Therefore, we believe that our work has provided novel insights with high scientific significance that are clear enough to make our work well distinguished from any other previously reported works, which should justify its suitability for publication in Nature Communications. But thanks to the reviewer's kind reminder, we have cited the paper (Adv. Mater. 2022, 34 (13), 2108419) in our manuscript.

Revisions to the manuscript:

“Notably, when the crystallite size scales down to the nanoscale, ceramic materials show high strength and large elastic strain endurance^{8,9,15-18}.”

Comment 1:

The recoverable strain > 10% mentioned in Abstract does not have a corresponding quantitative result in the main text.

Response:

We are thankful for the reviewer's comment. The recoverable strains were calculated by the pitch length changes of the BTO/CFO twist architectures. For example, based on the pitch length changes in Fig. 1c, the maximum recoverable strain is calculated as **26.8 %**, which is far beyond 10%. Additionally, we conservatively calculated recoverable strain based on the total length change of the structure and obtained the strain value of 8.3 %. After careful consideration, we decided to choose to report the total length change of the structure to avoid any misunderstanding. We have revised the manuscript as following.

Revisions to the manuscript:

“Here, we develop free-standing twisted architectures of nanoscale ferroic oxides showing shape-memory effect with a giant recoverable strain (>8 %).”

“In this work, we demonstrate shape-memory effect with giant recoverable deformations (>8 %) in freestanding architectures of ferroic oxide thin film by amplifying domain switching-induced strains through geometrical twist insertion.”

“When the tensile stress is sufficiently large, the structure maintained the deformation, as can be seen from the changed helical pitch length in Fig. 1c (recoverable strain calculated from the pitch length change is 26.8 % and from the total length change is 8.3 %).”

Comment 2:

The statement of “experimental access to the domain configuration unfeasible” lacks of factual basis, for example, the FIB-SEM has been used to reveal the domain configurations in BTO (10.1002/adma.202270103). Without the necessary microstructural characterization, the referee doubts the reliability of the followed simulation results. And, the influence of the defects and contamination introduced by the sample preparation on the ferroelectric domain switching are not clear.

Response:

We are grateful for the reviewer’s comment about the TEM measurements for the domain configuration analysis. As the reviewer mentioned, several different domain analyses of free-standing ferroelectric thin films have been reported (e.g., Science, 2019, 366 (6464), 475-479, Adv. Mater. 2022, 34 (13), 2108419, Sci. Adv. 2020, 6(34), eaba5847). In order to provide the reliability of our simulation results, we performed high-angle annular dark field scanning transmission electron microscope (HAADF-STEM) analysis on free-standing BTO/CFO membranes. We managed to make FIB lamella of free-standing BTO/CFO membranes with the curvature diameter of around 4.6 μm , in which the mechanical and electrical boundary conditions are almost the same as the twisted BTO/CFO (Fig. R2).

Fig. R2. High-angle annular dark field scanning transmission electron microscope (HAADF-STEM) analysis of free-standing BTO/CFO membrane. (a) Focused ion beam (FIB) lamella of the released BTO/CFO membrane. (b) HAADF-STEM image of the free-standing BTO/CFO layer. (c) Ti-ion displacements

have been mapped onto the HAADF-STEM images. In-plane ferroelectric polarization domains and the rotation of the polarization were observed in the BTO layer.

Ti-ion displacements in the BTO layer were analyzed and mapped onto the HAADF-STEM images to study the domain configurations. The BTO layer in the free-standing BTO/CFO showed not only in-plane domains because of the tensile strain imposed by the bottom CFO layer but also the rotations of the polarizations. Our simulation results are actually well in accordance with the previously reported simulation results (Fig. 4G in Science, 2019, 366 (6464), 475-479) and our experimental data also coincide well with polarization directions in the bent BTO films in the paper (Adv. Mater. 2022, 34 (13), 2108419, Science, 2019, 366 (6464), 475-479). Therefore, we believe that the microscopic ferroelectric domain analysis we provide here ensures the reliability of the phase field simulations presented in Fig. 3 of the main manuscript.

We provide the HAADF-STEM data and revised our manuscript as following.

Revisions to the manuscript:

“The BTO layer in the released BTO/CFO membrane exhibited ferroelectric properties with clear domain switching behavior (Fig. S2 and S3).”

Fig. S3. High-angle annular dark field scanning transmission electron microscope (HAADF-STEM) analysis of free-standing BTO/CFO membrane. (a) Focused ion beam (FIB) lamella of the released BTO/CFO membrane. (b) HAADF-STEM image of the free-standing BTO/CFO layer. (c) Ti-ion displacements have been mapped onto the HAADF-STEM images. In-plane ferroelectric polarization domains and the rotation of the polarization were observed in the BTO layer.

Comment 3:

The electron beam irradiation led to heating, electrostatic charging and vibration etc., it is unreasonable to directly attribute the increase of $E_{\text{dissipation}}$ to the irradiation-induced polarization. As shown in the SI videos, there was a significant charging at the top of the tip, and the effect of the complex electrostatic field on the beam induced-deformation is not clear too.

Response:

We are grateful to the reviewer's comment and pointing out the possibility of the effect of the heating, electrostatic charging, and vibration under scanning electron microscope (SEM). However, we believe that these effects are negligible and can be ruled out for the following reasons.

The effect of the heating by the electron beam are examinable since the heating effect strongly depends on the accumulated charges, i.e., the exposure time (ACS Nano 2020, 14, 4, 5053–5060; Adv. Func. Mater. 2019, 1902549). As reported in several papers (ACS Nano 2020, 14, 4, 5053–5060; J. Mater. Res, 2006, 21, 935-940; Phys. Rev. B, 2016, 94, 174104), the pure piezoelectric and electromechanical response from the ferroelectric polarization switching should only depend on the electric field or dose. In BTO/CFO twisted architectures, $E_{\text{dissipation}}$ value does not depend on the exposure time, but rather maintains constant value under the continuous electron beam exposure (after 4th cycle of Fig. 2e). Therefore, the effect of the heating can be ruled out for the reason of the shape memory effect in BTO/CFO twists.

In order to investigate the effect of the vibration and the electrostatic charging, we further measured the actuation force of the BTO/CFO twist (Fig. R3). The nanomechanical force sensor was attached to one end of the twist just like in Fig. 2a. Then, the force was measured at the initial (released; not stretched) state and at the stretched state of the BTO/CFO twist (black and red, respectively). At the released state, force change was not observed and the electron beam irradiation didn't have any effect on the force. In contrast, at the stretched state, the actuation (pulling) force was observed when the electron beam was irradiated onto the structure. If there is any electrostatic charging effect, the actuation force should be observed in BTO/CFO twist whether the structure is stretched or not. The fact that the actuation force was only observed when the BTO/CFO twist was stretched shows that the 'recovery' force is triggered only when there is a deformation. Therefore, the electrostatic charging effect can be ruled out. The vibration effect can be excluded as well. If there is any vibration generated by the electron beam exposure, the actuation force should depend on the beam scanning time. However, we have not observed any scanning time dependency, suggesting that the vibration effect on the recovery of the $E_{\text{dissipation}}$ is negligible.

Fig. R3. Actuation force triggered by the electron beam irradiation. Black and red lines indicate forces as a function of time while the BTO/CFO twist is at the initial (released) state or at the stretched state, respectively. When the BTO/CFO twist was not stretched (black), there was no response when the electron beam was irradiated to the structure. However, when the twist was stretched, the actuation force was measured when electron beam was irradiated onto the structure.

Additionally, we would like to further comment on the reviewer’s comment ‘beam induced-deformation’. In this report, we are showing permanent deformation caused by the external forces, such as mechanical stretching, electrostatic force (from the insulating substrate, Movie S5), and Van der Waals force, and eventually demonstrating the shape memory effect induced by the electron beam irradiation. Electron beam itself does not cause any ‘deformation’ in the original un-stretched BTO/CFO twists, which is different from the previously reported work.

We thus revised the manuscript to dispel possible doubts.

Revisions to the manuscript:

“The recovery of the $E_{dissipation}$ after the beam exposure and the beam dose dependency of the $E_{dissipation}$ clearly demonstrates the involvement of the polarization-switching modulated mechanism (Fig. S7). The other possible effects, such as heating, electrostatic charging, or vibration, can be ruled out. As can be seen in Fig. 2e, $E_{dissipation}$ does not depend on the exposure time and maintains constant value under the continuous electron beam exposure (after 4th cycle), indicating the shape memory effect is not induced by the heating^{10,39}. In addition, the actuation force measured with BTO/CFO twist proves that the electrostatic charging and vibration effect are negligible (Fig. S9).”

“In addition, the electric field-driven shape recovery of twisted architecture shows remarkable recoverable strains (Fig. 4b) as well as the clear actuation force (Fig. S9), providing opportunities to develop new type of the small-scale electromechanical actuating systems. The twisted architectures may also have the

potential to be actuated with other external stimuli, such as temperature and magnetic fields, since (i) BTO goes through phase transformations at Curie temperatures, and (ii) CFO is ferromagnetic, which may in turn broaden their applications.”

Fig. S9. Actuation force triggered by the electron beam irradiation. Actuation forces were measured by attaching nanomechanical force sensor at one end of the BTO/CFO twist. Black and red lines indicate forces measured as a function of time while the BTO/CFO twist was at the initial (released) state or at the stretched state, respectively. When the BTO/CFO twist was not stretched (black), there was no response when the electron beam was irradiated. However, when the twist was stretched, the actuation force was measured when electron beam was irradiated onto the structure. If there is any electrostatic charging effect, the actuation force should be observed in BTO/CFO twist whether the structure is stretched or not. The fact that the actuation force was only observed when the BTO/CFO twist was stretched show that the ‘recovery’ force is triggered only when there is a deformation. Therefore, the electrostatic charging effect can be ruled out. The vibration effect can be excluded as well. If there is any vibration generated by the electron beam exposure, the actuation force should depend on the beam scanning time. However, we have not observed any scanning time dependency, suggesting that the vibration effect on the recovery of the $E_{dissipation}$ is negligible.

Comment 4:

Phase-field model simulation was tried to unveil the physical mechanism, in which was a BTO/BTO structure. It is not consistent with the experimental BTO/CFO structure. The simulation results were too crude because there was not any quantitative result about the strain induced by the polarization. Therefore, I am confused about how the twisted geometrical design in the BTO/CFO bilayered thin films amplified the strain.

Response:

We are thankful for the reviewer's comment on the phase field simulation. However, we are afraid that reviewer might misunderstand our simulation process and results, and we are happy to take this opportunity to clarify the misunderstanding and re-stress our points.

First, as we described in detail in the Materials and Methods section, we performed phase-field model simulations on the small segment of the BTO layer in BTO/CFO freestanding structure, not the BTO/BTO structure. All the mechanical and electrical boundary conditions were considered for the BTO/CFO freestanding structure. For simplicity, we are only showing BTO slab in our results (Fig. 3) as CFO layer does not have any ferroelectric polarization and only gives mechanical constraints. Based on the reviewer's suggestions, we have revised the Fig. 3 and provided more quantitative data on the strain based on the phase-field model simulation.

Secondly, we would like to remind the reviewer that in our paper, the phase-field model simulation was used to investigate the strain and ferroelectric polarization change during the mechanical stretching and the electron beam irradiation (shape memory cycle), not to show the amplified strain in twisted geometrical design (again, different from the previously published paper Adv. Mater. 2022, 34 (13), 2108419). The amplification of the actuation strain and the shape memory effect can be ascribed to the strategic structural design, i.e., the twisted architecture. The introduction of the strategic geometrical designs to amplify the actuation strain is widely accepted technique in the field of shape memory alloys (Design of Shape Memory Alloy (SMA) Actuators, 2015, Springer Cham), and here we show the actuation strain amplification in ceramic materials for the first time using BTO/CFO twists. As previously reported (Nat. Mat., 2004, 3, 91-94), the shape memory effect by pure ferroelectric polarization switching is rather small ($< 1\%$). As we demonstrated with the finite-element method in the inset of the Fig. 2c, the whole structure could be elongated more than 20 % although the actual strain imposed in the materials by the stretching is quite small, which arises from the geometrical effect. Indeed, the giant recovery strain ($> 8\%$) in BTO/CFO twist total length can be attributed to the combination of the twisted geometrical design and the strain induced by the ferroelectric polarization switching.

We have revised our manuscript as following to avoid misunderstandings and make further clarification.

Revisions to the manuscript:

“Fig. 3a and 3b depict the in-plane strain and the ferroelectric domains in the BTO slab of freestanding BTO/CFO, respectively.”

“In the initial state, only the bottom surface was tensile-strained, representing strain coming from the CFO layer, with a domain configuration similar to that of the bent freestanding BTO films^{8,30} (top of Fig. 3a and Fig. 3b). When tensile stresses were applied on both sides of the slab (loading tensile force), the slab was mechanically stretched and the in-plane domain density increased (middle of Fig. 3a and Fig. 3b).”

“As negative charges accumulated on the slab surface during the electron beam exposure, ferroelectric polarization switched towards an out-of-plane orientation, giving rise to the change of strain profile and the retraction of the mechanically stretched BTO slab (bottom of Fig. 3a and Fig. 3b, quantitative description of

the effect of the tensile force and the surface charge accumulation on the shape recovery is provided in Fig. S8). The in-plane strain profile along the top surface of the BTO slab shows more clearly the increase of the tensile strain under stretching and the recovery under the change in the charge boundary condition (electron beam irradiation, Fig. 3c).”

Fig. 3. Phase field model simulation on the effect of mechanical stress and electron beam. (a) In-plane strain (ϵ_x) distribution and (b) in-plane ferroelectric polarization (P_x) mapping in the BTO slab calculated by the phase field model simulation. White arrows indicate the polarization directions. As tensile stress was applied in the BTO slab, the tensile strain and the in-plane domain density increased. The ferroelectric polarizations then switched to out-of-plane directions under surface electrical charging and it caused strain profile go back to the initial state, which can be seen in (c) in-plane strain profile of the top of the BTO slab. (d) Proposed cyclic behavior of the superelastic and shape-memory responses with schematic representation of the corresponding ferroelectric domain switching. The blue grids represent in-plane domains, and the red grids represent out-of-plane domains.

Reviewer 3:

General Comments:

In this work, the authors discover that the bilayer nanostructured $\text{BaTiO}_3/\text{CoFe}_2\text{O}_4$ coils possess superelasticity. Furthermore, they found the materials exhibit shape memory effect under electron irradiation after the coils are stretched to a high strain. A polarization-switching mechanism of ferroelectric layer is proposed to explain the superelasticity and shape memory effect. Overall, the results are quite interesting and publishable in NC although the shape memory effect and superelasticity have been reported in bulk and thin film ferroelectrics.

Response:

We are thankful for the reviewer's efforts in reviewing our manuscript. We are glad that the reviewer is favorably impressed and endorsed our work by mentioning "The results are quite interesting".

Comment 1:

The work lacks microstructure characterization. The authors attribute to the observed superelasticity and shape memory effect to the polarization-switching process. It implies that BTO is in the ferroelectric state. The authors may provide more results showing BTO is indeed in ferroelectric state. Also, if possible, the direct observation of the polarization or domain switching is also important.

Response:

Fig. R4. High-angle annular dark field scanning transmission electron microscope (HAADF-STEM) analysis of free-standing BTO/CFO membrane. (a) Focused ion beam (FIB) lamella of the released BTO/CFO membrane. (b) HAADF-STEM image of the free-standing BTO/CFO layer. (c) Ti-ion displacements have been mapped onto the HAADF-STEM images. In-plane ferroelectric polarization domains and the rotation of the polarization were observed in the BTO layer.

We are grateful for the reviewer's comment. Based on the reviewer's comment, we performed following characterizations of BTO/CFO free-standing structures to confirm the ferroelectric state of the BTO. We performed high-angle annular dark field scanning transmission electron microscope (HAADF-STEM) analysis on free-standing BTO/CFO membranes. Using focused ion beam technique, we produced the lamella structure of the free-standing BTO/CFO membranes (Fig. R4). Ti-ion displacements in the BTO layer were then analyzed and mapped onto the HAADF-STEM images to study the ferroelectric domain configurations. The BTO layer in the free-standing BTO/CFO showed not only in-plane domains because of the tensile strain imposed by the bottom CFO layer but also the rotations of the polarizations.

We also transferred the BTO/CFO free-standing membrane onto the Au-coated Si substrate and measured ferroelectric domain switching properties using piezoresponse force microscopy (PFM). After applying $V = \pm 10$ V, we could switch ferroelectric domains to out-of-plane up and down directions (Fig. R5). Additionally, local piezoelectric hysteresis loops show complete ferroelectric switching over electric voltage applications.

Fig. R5. Ferroelectric properties of free-standing BTO/CFO membranes. Ferroelectric properties were confirmed using piezoresponse force microscope (PFM). PFM (a) topography, (b) amplitude, and (c) phase images and local piezoelectric hysteresis (d) amplitude and (e) phase loops show ferroelectric domain switching properties of BTO in BTO/CFO.

We have added above microstructure characterizations of the BTO/CFO structures as following.

Revisions to the manuscript:

“The BTO layer in the released BTO/CFO membrane exhibited ferroelectric properties with clear domain switching behavior (Fig. S2 and S3).”

Fig. S3. High-angle annular dark field scanning transmission electron microscope (HAADF-STEM) analysis of free-standing BTO/CFO membrane. (a) Focused ion beam (FIB) lamella of the released BTO/CFO membrane. (b) HAADF-STEM image of the free-standing BTO/CFO layer. (c) Ti-ion displacements have been mapped onto the HAADF-STEM images. In-plane ferroelectric polarization domains and the rotation of the polarization were observed in the BTO layer.

Fig. S2. Ferroelectric properties of free-standing BTO/CFO membranes. Ferroelectric properties were confirmed using piezoresponse force microscope (PFM). PFM (a) topography, (b) amplitude, and (c) phase images and local piezoelectric hysteresis (d) amplitude and (e) phase loops clearly show ferroelectric domain switching properties of BTO layer in BTO/CFO.

Comment 2:

The film is composed of BaTiO_3 and CoFe_2O_4 . The authors simply attribute the observed superelasticity and shape memory effect to the ferroelectric layer. Does CoFe_2O_4 , which is also a ferroic material, contribute to the observe phenomena in the nanostructure? If not, evidence or discussion could be provided.

Response:

We appreciate for the reviewer's comment on the role of CoFe_2O_4 (CFO). It is well-known that the magnetic shape memory effect is originated from the magnetic field induced phase transformations that involve movement of the twin boundaries (ECS Transactions, 2007, 3 (25), 155-163; Adv. Func. Mat., 2009, 19, 983-998). However, CFO doesn't go through such phase transformation under magnetic fields, but only shows magnetostrictive properties (max. 600 ppm, Phys. Rev., 1955, 99, 1788). We believe that the magnetostriction of CFO itself is not enough to cause any shape memory effect in twisted BTO/CFO architectures based on our following experimental results.

Indeed, our CFO layer shows ferromagnetic behavior and its magnetic domain will react under the application of the magnetic fields (Fig. R5 a). In order to investigate the effect of the magnetostriction and/or the effect of the magnetic domain configuration changes in the CFO layer, we performed nanomechanical tensile tests under the external magnetic fields. To exclude the effect of the electron beam and the vibration as well as to apply magnetic fields, the setup was built on a vibration-isolated table equipped with optical microscope (Fig. R5 b and Movie S6). Although we have observed structural deformation under large tensile stress applications just like under the SEM, we have not observed any shape recovery with the magnetic field applications, up to 50 mT. In addition, $E_{\text{dissipation}}$ values during the tensile test cycles were also unaffected in the presence of magnetic fields (Fig. R5 c).

Fig. R5. Magnetic field dependency of the mechanical properties of BTO/CFO microhelices. (a) Magnetic hysteresis loop of the BTO/CFO/MgO (001) thin film. (b) In-situ nanomechanical measurement setup under optical microscope with magnetic field application system. The tensile test under optical microscope is available in Movie S6. Magnetic field was calibrated with magnetometer before applying the field. (c) Effect of the magnetic field (~50 mT) on the $E_{\text{dissipation}}$ over the tensile cycling tests. Unlike electron beam irradiation, no significant recovery was observed.

Revisions to the manuscript:

Based on the above results, we revised the manuscript as following:

“Still, the effects of the magnetostriction and/or the magnetic domain configurations in the CFO layer remain elusive. Other than imposing mechanical constraints in the BTO layer, the CFO layer could play a role in the shape memory effect through the magnetostriction or the change of the magnetic domain configurations. However, not only have we not observed any shape recovery, $E_{dissipation}$ values were also unaffected in the presence of magnetic fields, up to 50 mT (Fig. S10 and Movie S6).”

Fig. S10. Magnetic field dependency of the mechanical properties of BTO/CFO microhelices. (a) Magnetic hysteresis loop of the BTO/CFO/MgO (001) thin film. (b) In-situ nanomechanical measurement setup under optical microscope with magnetic field application system. The tensile test under optical microscope is available in Movie S6. Magnetic field was calibrated with magnetometer before applying the field. (c) Effect of the magnetic field (~50 mT) on the $E_{dissipation}$ over the tensile cycling tests. Unlike electron beam irradiation, no significant recovery was observed.

Comment 3:

From the results in the manuscript, shape memory is a real effect of the nanostructure. The elastic deformation of the coils is the amplified one due to the special structure of the material. The real strain of the material might not be large and be termed as superelasticity. The authors need prove that the superelasticity is also the property of the material.

Response:

We are grateful to the reviewer's comment on the superelasticity of BTO/CFO free-standing membranes. Superelasticity, the ability to deform to large strains recoverably, has been demonstrated in free-standing ferroelectric membranes such as BaTiO₃, SrTiO₃ and BiFeO₃ (Science, 2019, 366, 475-479; Nat. Commun., 2020, 11, 3141; Sci. Adv., 2020, 6, eaba5847). However, as reviewer mentioned, it is difficult to investigate the true superelasticity of the BTO/CFO membranes because of the contribution of the structural design (twisted architecture).

To resolve this issue, we measured the elastic strain and fracture strength of the BTO/CFO free-standing membranes by performing nanomechanical tensile tests (arxiv.org/abs/2210.09679). We designed dogbone-like free-standing BTO/CFO structures and obtained stress-strain curves by analyzing force-displacement curves. Compared to the bulk counterparts, BTO/CFO showed extremely large elastic strain (> 4 %), which proves the superelasticity of BTO/CFO structures. Furthermore, by transferring BTO/CFO layers to a stretchable polydimethylsiloxane (PDMS) substrate, we showed that the magnetoelectric coupling of the BTO/CFO layers can be reversibly tuned by applying and removing external tensile strain via stretching PDMS substrate (up to 4.1 % tensile strain). We believe that these results further corroborate the superelasticity of BTO/CFO free-standing membranes.

Revisions to the manuscript:

We have added following reference in our manuscript.

“Unlike brittle bulk ceramics, these freestanding twisted architectures exhibited a superelastic behavior¹⁹”

Comment 4:

The title is recommended to be “Shape-memory effect in twisted BaTiO₃/CoFe₂O₄ ferroic nanocomposites” because I am not sure the exact meaning of the word “giant” (“giant” strain?).

Response:

We are thankful for the reviewer's opinion on the title. Based on the comment, we changed the title to ‘Shape-memory effect in twisted BaTiO₃/CoFe₂O₄ ferroic nanocomposites’.

Revisions to the manuscript:

Title changed to

“Shape-memory effect in twisted BaTiO₃/CoFe₂O₄ ferroic nanocomposites”

REVIEWERS' COMMENTS

Reviewer #1 (Remarks to the Author):

The reviewer is very grateful to the authors for their effort to clarify the comments. Indeed, no shape memory phenomenon was reported in the free-standing BaTiO₃ [Science 366, 475-479 (2019) and ACS Nano 2020, 14, 4, 5053–5060]. The authors have fully addressed all my comments and the revised manuscript is essentially improved. Accordingly, I really believe this work is very significant and strongly suggest publication in Nature Communications.

Reviewer #2 (Remarks to the Author):

The authors have correctly responded all questions. I personally think this paper can be accepted for publication now.

Reviewer #3 (Remarks to the Author):

The authors have addressed the concern of the reviewers. I think it can accepted.

Responses to Reviewers' Comments

Reviewer 1:

The reviewer is very grateful to the authors for their effort to clarify the comments. Indeed, no shape memory phenomenon was reported in the free-standing BaTiO₃ [Science 366, 475-479 (2019) and ACS Nano 2020, 14, 4, 5053–5060]. The authors have fully addressed all my comments and the revised manuscript is essentially improved. Accordingly, I really believe this work is very significant and strongly suggest publication in Nature Communications.

Response 1:

We are grateful for the reviewer's positive comments and understanding of our contribution and his/her suggestions of acceptance.

Reviewer 2:

The authors have correctly responded all questions. I personally think this paper can be accepted for publication now.

Response 2:

We are grateful for the reviewer's efforts in reviewing our contribution and his/her suggestions of acceptance.

Reviewer 3:

The authors have addressed the concern of the reviewers. I think it can accepted.

Response 3:

We are grateful for the reviewer's positive comments and his/her suggestions of acceptance.